# Structure and Dynamics of Compact Dinucleosomes: Analysis by Electron Microscopy and spFRET

**DOI:** 10.3390/ijms241512127

**Published:** 2023-07-28

**Authors:** Maria E. Stefanova, Olesya I. Volokh, Oleg V. Chertkov, Grigory A. Armeev, Alexey K. Shaytan, Alexey V. Feofanov, Mikhail P. Kirpichnikov, Olga S. Sokolova, Vasily M. Studitsky

**Affiliations:** 1Biology Faculty, Lomonosov Moscow State University, Moscow 119234, Russia; durnopeyko.maria@gmail.com (M.E.S.); olesyavolokh@gmail.com (O.I.V.); chertkoff@gmail.com (O.V.C.); armeev@intbio.org (G.A.A.); shaytan_ak@mail.bio.msu.ru (A.K.S.); avfeofanov@yandex.ru (A.V.F.); sokolova@mail.bio.msu.ru (O.S.S.); vasily.studitsky@fccc.edu (V.M.S.); 2Shemyakin-Ovchinnikov Institute of Bioorganic Chemistry, Russian Academy of Sciences, Moscow 117997, Russia; 3Biological Faculty, MSU-BIT Shenzhen University, Shenzhen 518115, China; 4Fox Chase Cancer Center, Philadelphia, PA 19111, USA

**Keywords:** nucleosome, compact dinucleosome, dynamics, transmission electron microscopy, spFRET

## Abstract

Formation of compact dinucleosomes (CODIs) occurs after collision between adjacent nucleosomes at active regulatory DNA regions. Although CODIs are likely dynamic structures, their structural heterogeneity and dynamics were not systematically addressed. Here, single-particle Förster resonance energy transfer (spFRET) and electron microscopy were employed to study the structure and dynamics of CODIs. spFRET microscopy in solution and in gel revealed considerable uncoiling of nucleosomal DNA from the histone octamer in a fraction of CODIs, suggesting that at least one of the nucleosomes is destabilized in the presence of the adjacent closely positioned nucleosome. Accordingly, electron microscopy analysis suggests that up to 30 bp of nucleosomal DNA are involved in transient uncoiling/recoiling on the octamer. The more open and dynamic nucleosome structure in CODIs cannot be stabilized by histone chaperone Spt6. The data suggest that proper internucleosomal spacing is an important determinant of chromatin stability and support the possibility that CODIs could be intermediates of chromatin disruption.

## 1. Introduction

Nucleosomes are dynamic DNA–protein complexes; their positions along DNA and composition are modified by various factors including chromatin remodelers [1,2,3,4,5]. In particular, collisions between two adjacent nucleosomes occur and result in the formation of compact or overlapping dinucleosomes (CODIs) in yeast cells missing CHD1, ISW1 and/or ISW2 chromatin remodelers that are involved in the formation of properly spaced nucleosomes [6,7,8]. Formation of CODIs in the absence of the remodelers likely results in inhibition of transcript elongation by RNA polymerase II in vivo [8]. Formation of CODIs occurs during ATP-dependent chromatin remodeling by SWI/SNF in vitro [9,10]. Furthermore, nucleosome disassembly by chromatin remodelers proceeds through CODI formation in vitro [11] and depends on translocation activity of chromatin remodeler RSC (SWI/SNF family) [12]. Therefore, it is possible that CODIs are more dynamic complexes than mononucleosomes. However, structural dynamics of CODIs have not been studied previously.

The structures of various CODIs have been studied in vitro using DNA templates containing two overlapping high-affinity nucleosome-positioning sequences (NPSs) [10,13]. In these CODIs, DNA is partially uncoiled from one or both histone octamers, resulting in an overlap of the DNA regions previously occupied by the octamers [10]. In the case of a significant (54-bp) overlap between two NPSs, one histone H2A-H2B dimer was lost after formation of CODI [13]. Analysis of the crystal structure of an overlapping dinucleosome containing an octamer and a hexamer of histones has shown that the dinucleosome is a compact structure, in which a 250-bp DNA region is wrapped around the histones in three continuous DNA supercoils without a linker DNA between the DNA-bound octamer and hexamer [13].

Here we present an analysis of the structure and dynamics of CODIs containing a positioned nucleosome collided with an adjacent nucleosome formed on a natural DNA sequence. Our analysis using single-particle Förster resonance energy transfer (spFRET) microscopy in solution and in gel, as well as electron microscopy of purified CODIs suggests that both nucleosomes within the CODI are dynamic structures. The observed dynamics involve transient uncoiling of nucleosomal DNA from the surface of histone octamer facing adjacent nucleosome. As a result, histone chaperone Spt6 does not affect this more open nucleosome structure.

## 2. Results

### 2.1. The Primary Experimental Approach

In previous structural studies, the structure and composition of dinucleosomes were mostly dictated by the spacing between two strong NPSs on the DNA that is largely dependent on the H3/H4 histone tetramer-binding regions of NPS [10,13]. However, spontaneous or remodeler-facilitated nucleosome collisions on natural DNA sequences could result in a different outcome. Therefore, in our experiments the DNA template for CODI formation contained a high-affinity 147 bp 603 NPS [14] that was positioned next to a natural 122-bp DNA sequence. The 603 DNA sequence directs precise nucleosome positioning [15,16], while the 122-bp DNA sequence provides a DNA region that is short enough to provide space constraint and enforce CODI formation.

The DNA template was labeled with a pair of fluorophores (donor Cy3 and acceptor Cy5) localized within the 603 NPS. Mono- (1 × N) and dinucleosomes (2 × N) were assembled on the DNA template in vitro using H1-free chromatin from chicken erythrocytes, gel-purified and analyzed by the native polyacrylamide gel electrophoresis (PAGE). Both gel-purified complexes were about 90% pure and contained minor amounts of free DNA (Figure 1). In the nucleosome assembled on 603 NPS, the fluorescent labels are localized close to each other on the neighboring gyres of nucleosomal DNA (Figure 1). This supports an efficient Förster resonance energy transfer (FRET) and allows monitoring of changes in the nucleosome structure by measuring proximity ratio E_PR_, i.e., FRET efficiency without correction for quantum yields of labels and an instrumental factor [17,18,19,20]. The positions of the labels were selected based on the known structure of the nucleosome [21] in such a way that the labels were oriented outward and did not disturb DNA–histone interactions necessary for proper nucleosome assembly.

In total, three different positions of fluorescent labels were used (Figure 2a). For the pairs of labels attached to nucleotides in the positions 13 and 91 bp or to 57 and 135 bp from the boundary of 603 NPS, one of the labels in a pair was localized close to the boundary of the 603 NPS (Figure 2a). The labels attached to 35 and 112 bp were positioned further in the nucleosome (Figure 2a). With these combinations of the fluorescent labels the alterations in the distances between DNA gyres of the 603 nucleosome can be revealed within 1 × N and 2 × N using spFRET microscopy (Figure 1). The efficiency of the energy transfer from the donor to acceptor is higher when the distance between labels is shorter, and decreases with an increase in the distance. Accordingly, spFRET can detect both DNA uncoiling from the histone octamer and unfolding of nucleosomal DNA together with the associated histones [17,22].

spFRET microscopy in solution is performed at a very low concentration of nucleosomes, when no more than one nucleosome diffuses through the focus of a laser beam at any given time and contributes to the measured signal. Such an approach allows probing a structure of each measured nucleosome and revealing subpopulations of nucleosomes with different structural features. spFRET microscopy in gel allows FRET measurements from biochemically homogeneous supramolecular complexes, thus providing clues on their intrinsic dynamics. Similarly, electron microscopy provides images of structurally different classes of supramolecular complexes. Finally, molecular modeling allows interpretation of the observed structural difference on a molecular level and provides important clues on the possible mechanism of the observed conformational transitions.

### 2.2. Nucleosomal DNA Is Partially Uncoiled in Dinucleosomes

To compare the structures of the same nucleosomal DNA within the mononucleosomes and CODIs, corresponding gel-purified complexes were analyzed using spFRET microscopy in solution. Mononucleosomes are characterized by a typical frequency distribution in terms of E_PR_ measured with spFRET microscopy that is described by two Gaussian peaks (Figure 2b–d): the minor peak (low E_PR_; mean E_PR_ of 0.02 ± 0.02) that corresponds to nucleosomes with a longer distance between labels and a major peak (high E_PR_; mean E_PR_ of 0.65 ± 0.02, typically >85% of the individual signals) that corresponds to nucleosomes with a shorter distance between labels. The peaks correspond to intact nucleosomes (high E_PR_) and free DNA and/or nucleosomes with partially uncoiled DNA (low E_PR_). Similar E_PR_ profiles have been described previously [17,18,19,20] and were characteristic for mononucleosomes containing the labels at different positions on NPS DNA: 13/91, 35/112 and 57/135 bp (Figure 2). Nucleosomes labeled at 13/91 and 57/135 bp contained a higher fraction of nucleosomes with low E_PR_ in comparison with nucleosomes labeled at 35/112 bp, suggesting increased nucleosome breathing (partial and temporary uncoiling of DNA) at the boundary regions of a nucleosome [17,18,22].

E_PR_-profiles for the mononucleosomes and dinucleosomes labeled either at 35/112 bp or at 57/135 bp in solution are similar (Figure 2c,d). In contrast, a significant difference was detected between mono- and dinucleosomes labeled at 13/91 bp, i.e., close to the nucleosome boundary facing the adjacent nucleosome (Figure 2b). Subpopulations of mononucleosomes and dinucleosomes with low E_PR_ values were 9.3 ± 0.3% and 24 ± 2% (*p* = 0.0002), respectively (Figure 2b). This suggests that the presence of the adjacent histone octamer results in partial uncoiling of nucleosomal DNA from the positioned nucleosome at the boundary facing the adjacent nucleosome. The increased subpopulation of low-FRET nucleosomes observed for CODI as compared with mononucleosomes reflects an appearance of an additional conformation of CODI, which is either a mixture of low- and high-FRET stable states or a low-FRET state that is in a dynamic equilibrium with the high-FRET conformation.

### 2.3. Multiple Conformations of CODIs Are Present within a Single Band on a Native Gel: spFRET in-Gel (spFRETG) Approach

To discriminate between the stable states and dynamic equilibrium possibilities, bulk FRET was measured for the mono- and dinucleosomes in a non-denaturing gel (Figure 3a). It was expected that CODIs bearing considerably different conformations migrate differently in the gel. However, both mono- and dinucleosomes labeled at 13/91 bp migrate in the gel as single discrete bands and have different FRET (observed as different colors of the bands in Figure 3a), indicating that the partial uncoiling of the nucleosomal DNA detected by spFRET in solution is also evident in the bulk population of CODIs. Since CODIs that are characterized by considerably different FRET efficiencies (low and high E_PR_, respectively, Figure 2b) migrate as a single band (Figure 3a), they probably exist as a dynamic mixture of the “open” and “closed” complexes (Figure 2a) moving as a single band in the non-denaturing gel.

To evaluate the possibility that electrophoresis itself affects the equilibrium between the different conformations of the CODI, the complexes were separated by native PAGE and analyzed using spFRET in-gel (spFRETG) approach (Figure 3b). Since the spFRETG approach was developed and applied for the first time, it was initially validated with the well-studied mononucleosomes [23] and their complexes with protein complex FACT [17]. Mononucleosomes in gel are characterized by the typical E_PR_-profile with two peaks that are very similar in positions and relative intensities to those observed in the E_PR_-profile of mononucleosomes in solution (Appendix A). The large-scale unwinding of nucleosomal DNA by yeast FACT was similarly observed in the E_PR_-profiles of FACT-mononucleosome complexes both in solution and in gel (Appendix A). The data suggest that native electrophoresis minimally affects the conformation of nucleosomes and other supramolecular complexes, and FRET characteristics of the Cy3/Cy5 fluorophores are not noticeably affected by the electrophoresis in the gel.

Accordingly, the E_PR_-profiles of CODIs were compared in solution and in gel and found to be similar (Figure 3c and Appendix A). The data suggest that considerably different multiple conformations of CODIs are present in one band on the native gel; the dynamic equilibrium between these conformers likely precludes their separation during electrophoresis.

### 2.4. Heterogeneity in the Structure of CODIs: Analysis by Electron Microscopy

To further investigate the structural heterogeneity of CODIs, gel-purified complexes (Figure 1) were analyzed using electron microscopy (Figure 4a,b). This analysis revealed that the CODIs have variable structures with distances between the particles ranging from ~11–27 nm (Figure 4b). In the majority of the complexes, distances between the centers of particles are changed stepwise, with 3–4 nm increments (approximately one turn of the double helix, 3.6 nm), corresponding to distances of ~11, 15 and 18 nm (Figure 4b). Since the length of the DNA helical repeat (10 bp) is 3.6 nm, the data suggest that DNA in the dinucleosome is uncoiled from the histone octamer stepwise, approximately one turn of the DNA helix at a time.

When a larger distance between the particles was observed in the electron microscopy images, one nucleosome projection was typically smaller than the other (Figure 4a) indicating that DNA is more uncoiled from one of the octamers. The data suggest that histone content in one of the two nucleosomes was changed, possibly due to transient loss of an H2A/H2B dimer from the nucleosome containing uncoiled DNA. Alternatively, relative orientation of the nucleosomes in the complexes (Figure 4a) is changed as the distance between them is increased. In combination with the results of spFRET analysis, the data suggest that CODIs contain nucleosomes exhibiting variable structures.

### 2.5. Modeling Structure and Dynamics of the Dinucleosome

To further evaluate the structures of CODIs, a molecular modeling was conducted. In our model of the dinucleosome, the distance between nucleosomes is described in terms of three parameters: the length of the linker DNA (in bp) between two NCs (further termed “offset”) and the lengths of DNA fragments that were uncoiled from the first and second NCs, (see Appendix A). Uncoiled nucleosomal DNA was modelled with straight segments of B-DNA.

The length of nucleosomal DNA that can be uncoiled in dinucleosomes was estimated using the data obtained with atomic force microscopy [10]. The study of dinucleosomes formed on the DNA template, which contained two NPSs separated by a DNA sequence of variable length (offset of zero or 48 bp), revealed that internucleosomal distances were longer than the introduced offsets [10]. Since nucleosomes are formed on identical NPSs, uncoiling of nucleosomal DNA likely occurs symmetrically from both nucleosomes. Our modeling shows that the length of the uncoiled DNA in these dinucleosomes is not dependent on the offset value (zero or 48 bp), but depends on the presence of divalent ions in solution and is equal to 17 and 35 bp per a nucleosome in the presence and absence of divalent ions, respectively (Appendix A). The presence of states with similar uncoiling of DNA from the histone octamer in mononucleosomes was shown by cryo-electron microscopy [24].

Our experimental conditions were very similar to those described in [10] in the presence of divalent ions. Accordingly, in the first approximation, we assumed that the length of uncoiled nucleosomal DNA in our CODIs is also equal to 17 bp per nucleosome. Taking into account the distances between nucleosomes revealed in our electron microscopy experiments, the offsets of −7, 4 and 15 bp were calculated for the CODI structures “1”, “2” and “3” observed by electron microscopy, respectively (Figure 4a,b); a negative offset indicates that nucleosome territories are overlapped.

Our assumption about the 17 bp DNA uncoiling per nucleosome allows one to describe the structural states of dinucleosomes observed by electron microscopy. It is also consistent with spFRET microscopy data about high FRET efficiency in the major subpopulation of CODIs (Figure 2 and Figure 3), because uncoiling of 17 bp of nucleosomal DNA from nucleosomes labelled at 13/91, 35/112 or 57/135 bp does not induce a noticeable decrease in E_PR_ (17).

However, this assumption does not allow one to explain the low FRET efficiency in a minor subpopulation of CODIs labeled at 13/91 bp (Figure 2 and Figure 3). We have previously shown that E_PR_ < 0.333 is expected for nucleosomes labeled at 13/91 bp when the uncoiling of nucleosomal DNA exceeds 30 bp (17). Therefore, DNA uncoiling in CODIs having low FRET efficiency likely occurs asymmetrically: at least 30 bp of nucleosomal DNA are uncoiled from the nucleosome formed at NPS, and only minimal or no DNA uncoiling occurs in the second nucleosome (Appendix A, bottom). Indeed, the internucleosomal distances in the models with asymmetric and symmetric uncoiling of nucleosomal DNA are similar and the offset is maintained.

The proposed model of extensive asymmetric uncoiling of nucleosomal DNA in the dinucleosome predicts a considerable destabilization of the nucleosome formed on the NPS.

### 2.6. DNA Uncoiling Is Differently Affected in Mono- and Dinucleosomes by Histone Chaperone Spt6

Previously, we have shown that the human histone chaperone FACT hinders DNA uncoiling in mononucleosomes, thus stabilizing their structure [18]. Here, we studied whether another histone chaperone, yeast protein Spt6, which has overlapping functions with FACT complex [25,26,27], can affect uncoiling of nucleosomal DNA. Like FACT, Spt6 is involved in transcript elongation, and can bind DNA, nucleosomes, histone H2A-H2B dimer and H3-H4 tetramer [25,28].

According to spFRET analysis, Spt6 protein stabilizes mononucleosomes: the fraction of mononucleosomes containing uncoiled nucleosomal DNA is decreased 2.8 fold in the presence of Spt6 as compared to mononucleosomes incubated in the absence of Spt6 (Figure 5a,c). However, Spt6 cannot significantly stabilize the CODIs: the fraction of CODIs containing uncoiled nucleosomal DNA was reduced only 1.35 times in the presence of Spt6 (Figure 5b,c). The data suggest that in the case of colliding nucleosomes, the histone chaperone Spt6 minimally affects the structural dynamics of dinucleosomes and cannot correct the observed uncoiling of nucleosomal DNA.

## 3. Discussion

In summary, spFRET data suggest that the partial, reversible and limited uncoiling of nucleosomal DNA from at least one of the nucleosomes occurs in ~15% of CODIs (Figure 2 and Figure 3). Dinucleosomes migrate as one band in the native gel, but are characterized by different FRET efficiencies and different distances between nucleosomes (Figure 3 and Figure 4), suggesting that CODIs are more structurally dynamic than mononucleosomes. Finally, dinucleosomes are less prone to stabilization by a histone chaperone (Figure 5). Taken together, the data suggest that CODIs are more open and less stable than mononucleosomes, even in the presence of histone chaperones.

Our spFRET data show that the number of particles with low FRET efficiency increases in CODIs as compared to mononucleosomes. Our modeling estimates suggest that in our experimental setup the nucleosomal DNA has to be uncoiled by more than 30 bp in order to explain the observed low FRET efficiency. Taking into account the data on the extent of nucleosomal DNA uncoiling in the dinucleosome, where nucleosomes were separated by linker DNA of 0 to 48 bp length [10], our modelling analysis of CODIs with colliding nucleosomes (Appendix A) predicts the following. Binding of the second histone octamer on the DNA template having a length, which is insufficient to accommodate two complete nucleosomes, results in the invasion of the second nucleosome into the territory of the first nucleosome accompanied by uncoiling of nucleosomal DNA. The two colliding nucleosomes would likely be involved in a dynamic interaction facilitated by the long N-terminal tails of histone H3 [29].

Our modeling of the structure of colliding nucleosomes (Figure 4c, top) suggests ~7 bp overlap between the territories of the two nucleosomes. Recent cryo-electron microscopy data support our model of dinucleosomes as dynamic entities with transiently and asymmetrically unwrapped DNA [24]. These asymmetrical dynamics likely induce the transition between the high and the low FRET states (Figure 2a).

Previously, it was shown that the distance between mononucleosomes in various dinucleosomes is larger than the expected length of the linker DNA segment connecting histone octamers or the octamer and hexamers (in the case of CODIs) [10]. Accordingly, it was proposed that DNA uncoiling occurs more efficiently in component nucleosomes of a dinucleosome than its uncoiling in isolated mononucleosomes [10]. Our electron microscopy data are consistent with this proposal, and, in combination with the spFRET data, suggest that the collision of nucleosomes induces partial uncoiling of a considerable (~30 bp long) DNA fragment from the histone octamer.

Computational analysis of dinucleosome dynamics in the presence of an external force showed asymmetric uncoiling, when the length of linker DNA is short (less than 10 bp) [30]. Notably, our experimental data show asymmetric uncoiling of CODIs, suggesting that compact dinucleosomes possess some of the features of the dinucleosomes separated by a linker DNA.

Is the partial uncoiling of DNA in dinucleosomes reversible? Previous structural studies have shown that formation of CODIs where octamer territories overlap by 44 or more base pairs is accompanied by the loss of one H2A/H2B dimer [10,13]. Similarly, a loss of H2A/H2B dimer was observed in mononucleosomes missing more than 30-bp DNA region [24]. Since in our case maximal combined uncoiling from both adjacent nucleosomes involves a 30-bp DNA region, and because one of the nucleosomes occupies only 122-bp DNA region, some H2A/H2B dimer loss could be expected from this, but not for the second nucleosome. However, a dinucleosome missing one H2A/H2B dimer is expected to have a considerably different mobility in a native gel, while in our case no alternative bands on the gel were observed after gel purification of the dinucleosomes. Accordingly, our data suggest that when octamer territories are minimally overlapped or not overlapped, histones are not lost from either nucleosome in the dinucleosome.

Multiple direct in vitro experiments suggest that CODIs are formed by ATP-dependent complexes that belong to the SWI/SNF family [9,10,11,31,32]. According to micrococcal nuclease analysis of human chromatin, overlapping dinucleosomes are formed in the region downstream of transcription start site [13]. Functions of SWI/SNF remodelers and RNA-polymerases in this region are supported by histone chaperones FACT and Spt6 [33,34]. FACT has been suggested to bind CODIs: ChIP-seq of SPT16 after digestion by micrococcal nuclease showed that FACT binds DNA fragments of the compact dinucleosome size 150–300 bp [35,36]. We speculate that ATP-dependent remodelers could induce collisions of nucleosomes that in turn compromises their structures and facilitates removal of some nucleosomes from the binding sites of various regulatory factors. Histone chaperone Spt6 does not strongly interfere with the CODIs structure, but after removal of one of the two nucleosomes could stabilize the remaining one. It is possible that other histone chaperons can affect CODIs differently. For example, FACT exhibits nucleosome destabilization activity [17,22] and possibly destabilizes CODIs near transcription start sites to facilitate transcription.

How could our model of dynamic dinucleosomes work in the context of the chromatin structure within the cell nuclei? In a continuous chain of nucleosomes, there are no DNA ends that stop nucleosome sliding along DNA in vitro. Therefore, in a chain of nucleosomes two effects of colliding nucleosomes are possible: DNA uncoiling and/or sliding of core histones to a new position. If sliding is prevented by some factors, the preferred mechanism would be DNA uncoiling. If, however, sliding is facilitated by some factors (such as chromatin remodelers Isw1 and Chd1), sliding would be observed.

## 4. Materials and Methods

### 4.1. Proteins and Nucleosomes

Mono- and dinucleosomes were assembled by octamer transfer from chicken erythrocyte donor -H1 chromatin after dialysis from 1M NaCl and gel-purified as described earlier [17,20,37]. Nucleosomal DNA was made by polymerase chain reaction (PCR) using the following fluorescently labeled primers: Mid_Fw 5′—AGCGACACCGGCACTGGGCCCGGTTCGCGCTCCCTCCTTCCGTGTGTTGTCGTCTCT CGGGCGTCTAAGTACGCT*T (where T*—nucleotide, labeled by Cy3), Mid_Rev 5′—ACCCCAGGGACTT^#^GAAGTAATAAGGAGAGGGCCTCTTTCAACATCGATGCACG GTGGTTAGCCTTGGA (where T^#^—nucleotide, labeled by Cy5). Promoter DNA was obtained by PCR using the following primers: Fw 5′—CCGGGATCCAGATCCCGAAAATTTA, Rev 5′—CGCGAACTGGGCCCCAGTGCCGGTGTCGCTTGGGTTGGCT. The pDS plasmid carrying an insert with the nucleosome positioning sequence 603 and the T7A1 promoter was used as a template for PCR. Final matrix sequence: CCGGGATCCAGATCCCGAAAATTTATCAAAAAGAGTATTGACTTAAAGTCTAACCTATAGGTACTTACAGCCATCGAGAGGGACACGGCGAAAAGCCAACCCAAGCGACACCGGCACTGGGCCCGGTTCGCGCGCCCGCCTTCCGTGTGTTGTCGTCTCTCGGGCGTCTAAGTACGCTTAGCGCACGGTAGAGCGCAATCCAAGGCTAACCACCGTGCATCGATGTTGAAAGAGGCCCTCCGTCCTTATTACTTCAAGTCCCTGGGGT.

Promoter and (Cy3, Cy5)-labeled nucleosomal DNA were incubated with TspRI enzyme (New England Biolabs, Ipswich, MA, USA) in NEB4 buffer (New England Biolabs) for 3 h at 65 °C. The resulting DNA fragments were separated by electrophoresis (4–6 V/cm, 2 h) in 1.5% agarose gel in 0.5× TBE buffer (44.5 mM Tris, 44.5 mM boric acid, 1 mM EDTA) with addition of 4 M urea and 0.5 μM ethidium bromide.

Ligation of 1–2 µg of purified promoter and (Cy3, Cy5)-labeled nucleosomal DNA fragments was performed at a molar ratio of fragments of 1.15:1 in T4 DNA ligase buffer (15 h, 16 °C). Separation of the product and unligated fragments was carried out in a 1.8% agarose gel in 0.5× TBE buffer containing 0.5 mg/mL ethidium bromide at 4–6 V/cm for 2 h. DNA was purified from the gel using DNA extraction kit (Omega Bio-Tek kit, Singapore) following the manufacturer’s instructions. DNA concentration was determined by measuring the optical density at a wavelength of 260 nm.

As a source of histones for the assembly of nucleosomes, we used chromatin without histone H1, which was isolated from chicken erythrocytes as described previously [38]. Nucleosomes were assembled on the (Cy3, Cy5)-labeled DNA template during dialysis against decreasing NaCl concentration (1 M, 0.75 M, 0.5 M and 10 mM NaCl) according to the protocol described in [38].

For gel purification of mononucleosomes and dinucleosomes 4.5% PAAG was used. Pre-electrophoresis was carried out at 200 V for 2 h in HE buffer (10 mM HEPES-NaOH, pH 8.0, 0.2 mM EDTA). Then, the buffer in the cameras was replaced with a fresh one and pre-electrophoresis was performed for another 30 min. A sample containing mono- and dinucleosomes was loaded on the gel after adding up to 10% sucrose to the sample. Electrophoresis was carried out for 2–2.5 h at constant current (5 mA) at 4 °C. The position of Cy5-labeled mono- and dinucleosomes in the gel was detected by scanning the gel with a Typhoon (GE Healthcare, Chicago, IL, USA). The gel bands containing mono- and dinucleosomes were cut out. Then, the gel section containing the band was extracted in HE/BSA buffer (10 mM HEPES-NaOH, pH 8.0, 0.2 mM EDTA, 200 mg/mL BSA) that was added to the crashed gel at a 1:1 volume ratio of buffer:gel. The sample was left for 15 h at 4 °C. Next, another 50–100 μL of HE/BSA buffer was added to the sample, The gel was precipitated by centrifugation and the supernatant containing mono- or dinucleosomes was taken. Mononucleosomes and dinucleosomes were stored at 4 °C.

Spt6 was purified as described [25]. Nhp6 was expressed in *Escherichia coli* and purified as described [39,40]. Spt16/Pob3 was purified as the intact heterodimer from the yeast cells overexpressing both proteins [41,42].

### 4.2. spFRET Measurements in Solution

spFRET measurements of nucleosomes in solution were performed at the ~1 nM concentration in the buffer 20 mM Tris–HCl (pH 8.0), 5 mM MgCl_2_, 2 mM 2-mercaptoethanol and 40 mM KCl as described [17,20]. The duration of each of two successive measurements was 10 min, and the number of measured nucleosomes was 2000–5000 particles.

Measurements were performed using an LSM710-Confocor3 laser scanning confocal microscope (Zeiss, Oberkochen, Germany) with 40× C-Apochromat water immersion objective (NA 1.2) in 8-well chambers on a Lab-Tek cover slip (Thermo Scientific, Waltham, MA, USA). Fluorescence was excited by an Ar+ ion laser (514.5 nm, 2 μW under the objective) and recorded using avalanche photodiodes in the ranges of 530–635 nm (Cy3) and 635–800 nm (Cy5). The diameter of the confocal diaphragm was equal to 1 Airy disk. For each sample, the dependencies of the fluorescence intensity on time were measured for 15 min with an integration constant of 5 ms. The analysis included nucleosomes with signal intensity I3 = 10 ÷ 80 kHz and I5 = 5 ÷ 80 kHz, where I3 and I5 are the signal intensities of Cy3 and Cy5. I3 and I5 were corrected for a background value of 1.0 and 0.5 kHz, respectively. Cy3 and Cy5 fluorescence intensities measured for each nucleosome were converted into FRET efficiency (E) according to the equation:E= (I5 − 0.19 × I3)/(I5 + 0.81 × I3)(1)
where the coefficients 0.19 and 0.81 are introduced to account for the partial overlap of the fluorescence spectra of Cy3 and Cy5 in the 635–800 nm region. The set of E values was graphically represented as the relative frequency distribution of the value of E and described as a superposition of Gaussian bands. GraphPad Prism software (version 5.0) was used for statistical calculation of the results.

### 4.3. FRET and spFRET Measurements in Gel

Native PAGE and bulk FRET measurements in gel were performed as described [23]. spFRET measurements in gel were carried out as follows. Nucleosomes were separated by native PAGE and scanned with the Amersham Typhoon RGB laser scanner (GE Healthcare Bio-Sciences AB, Uppsala, Sweden) to reveal positions of bands corresponding to mononucleosomes and dinucleosomes. Gel was placed between object and cover glasses and analyzed with the confocal microscope using the same optics and parameters as for the spFRET measurements in solution [17,20]. The areas of measurement were selected at the edges of the analyzed bands, where the concentration of nucleosomes was low enough to allow the analysis of individual particles. The position of the laser focus shifted along the band by 10 µm each 5–10 s during the 10-min measurement period. Further treatment and presentation of the obtained data was conducted in the same way as for nucleosomes in solution [17,20].

### 4.4. Electron Microscopy

Dinucleosomes were gel-purified, transferred to the buffer containing 20 mM Tris–HCl (pH 8.0), 5 mM MgCl_2_, 2 mM 2-mercaptoethanol, and 40 mM KCl, and analyzed using single-particle electron microscopy after negative staining as described [22]. Images were obtained with a JEM2100 microscope (JEOL, Tokyo, Japan) at ×40,000 magnification (Appendix A). Subsequently, ~25% of single particles in the dinucleosome preparation were similar in size to mononucleosomes (Appendix A and see the legend to Figure 5).

A total of 11,944 dinucleosomal particles were automatically collected and classified into 50 2D classes using CryoSparc [43]. The 16 best 2D classes were selected that contained 4956 particles. The distances between the centers of histone octamers in all selected particles were then measured using ImageJ program (National Institutes of Health, Bethesda, MD, USA).

### 4.5. Incubation of Nucleosomes in the Presence of Histone Chaperones

Purified mono- and dinucleosomes were incubated with Spt6 (0.1 µM) for 10 min at 30 °C in buffer containing 20 mM Tris–HCl (pH 8.0), 5 mM MgCl_2_, 2 mM 2-mercaptoethanol, 150 mM KCl.

For spFRET measurements in gel, complexes of yeast FACT with labeled nucleosomes were formed in solution containing 17 mM HEPES pH 7.6, 2 mM Tris-HCl pH 7.5, 0.8 mM Na_3_EDTA, 0.11 mM 2-mercaptoethanol, 11 mM NaCl, 1.1% glycerin, 12% sucrose (10 min at 30 °C) and evaluated by the electrophoretic mobility shift assay as described [44]. Concentrations of nucleosomes, FACT (protein complex Spt16/Pob3) and Nhp6 were 0.5 nM, 0.13 µM and 1.3 µM, respectively. spFRET measurements of yeast FACT complexes with nucleosomes in solution were conducted in the same buffer.

### 4.6. Modeling Dinucleosome Structures

Dinucleosome structures were built using reference structure with pdb code 3lz0. Unwrapped DNA and additional DNA between nucleosomes was modeled with straight B-DNA using 3DNA software (version 2.1). Histones from the reference structure were fitted to both nucleosomes by minimizing the RMSD for nucleotides in the dyad region using only sugar phosphate atoms. Internucleosome distances were calculated between the centers of geometry of the histone octamers. The changes in E_PR_ for the obtained models were assessed from the distances between the labeled nucleotides using formula E_PR_ = 1/(1 + (D/R_0_)^6^), where D is the distance between nucleotide phosphates and R_0_—Forster radii for Cy3/Cy5 labels (~5.5 nm).

## Figures and Tables

**Figure 1 ijms-24-12127-f001:**
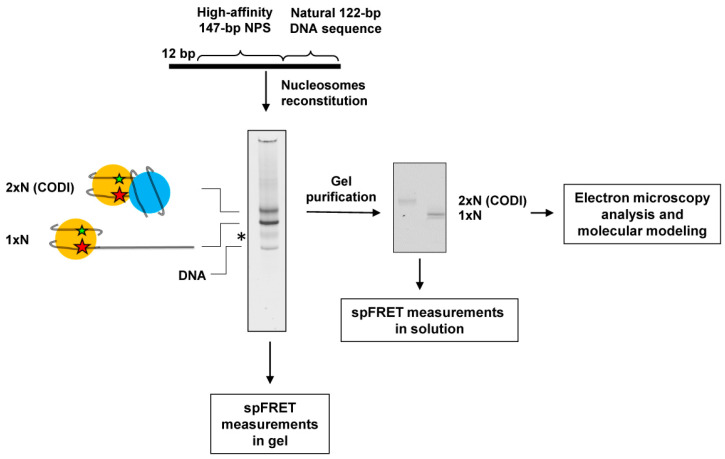
The primary experimental approach. Mono- (1 × N) and dinucleosomes (2 × N) were assembled on the DNA template, containing 603 NPS and adjacent natural DNA sequence. Fluorescent labels (donor Cy3 and acceptor Cy5, green and red asterisks, respectively) were introduced into neighboring gyres of the 603 nucleosomal DNA. Mono- and dinucleosomes were analyzed by native PAGE (on the left); black asterisk shows the position of a minor contaminating band of unknown origin. Gel-purified 1 × N and 2 × N (in the middle) were used for spFRET and electron microscopy experiments. Data obtained from spFRET and electron microscopy were used in molecular modeling experiments to create a structural model of the dinucleosome.

**Figure 2 ijms-24-12127-f002:**
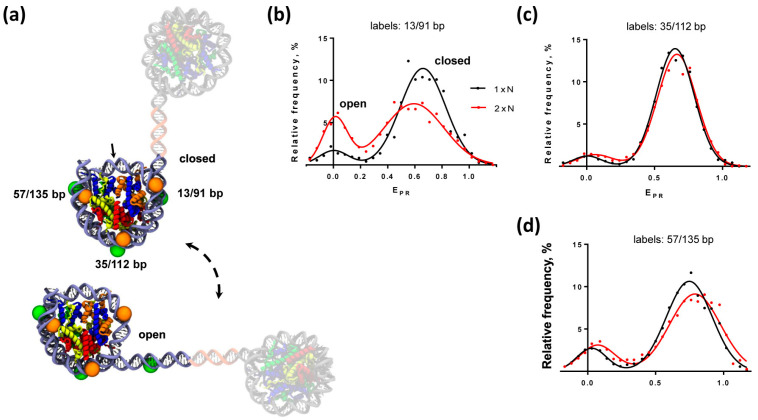
Nucleosomal DNA is partially uncoiled from the histone octamer in dinucleosomes. (**a**) Positions of different pairs of the fluorescent labels Cy3 (green circle) and Cy5 (orange circle) on nucleosomal DNA in the closed and open dinucleosomes. The nucleosomal dyad is shown by the arrow. (**b**–**d**) Analysis of mono- and dinucleosomes by spFRET in solution. Typical frequency distributions for mono- (black curves) and dinucleosomes (red curves) by the proximity ratio (E_PR_) are shown. Nucleosomes were labeled at 13/91 bp (**b**), 35/112 bp (**c**) and 57/135 bp (**d**).

**Figure 3 ijms-24-12127-f003:**
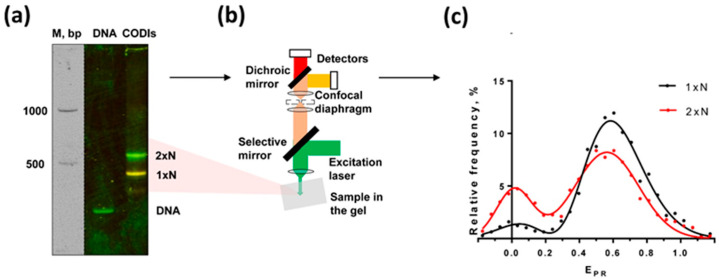
Multiple conformations of the dinucleosomes are present within a single band on a native gel: spFRET in-gel approach. (**a**) Characterization of mono- (1 × N) and dinucleosomes (2 × N), which were labeled at 13/91 bp, by in-gel FRET. Complexes were separated by native PAGE, and the gel was analyzed as described in Methods. Unfolding of nucleosomal DNA was detected as a decrease in FRET efficiency (change in color from yellow to green). (**b**) Experimental approach for spFRET measurements within single bands on the native gel. (**c**) The fragment of gel containing nucleosomes and CODIs was analyzed using spFRET. Designations as in Figure 2b.

**Figure 4 ijms-24-12127-f004:**
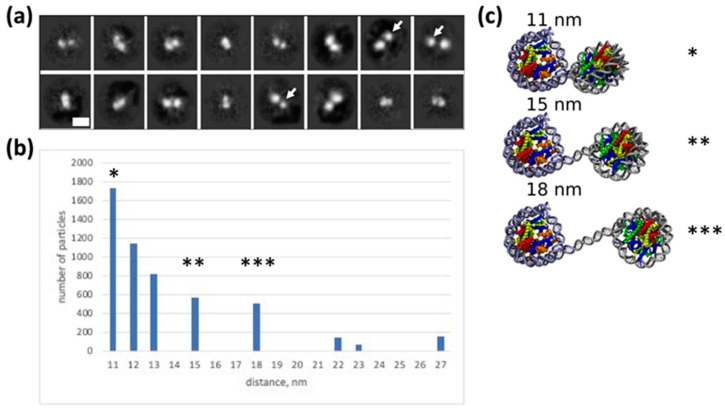
Heterogeneity in the structure of CODIs: analysis by electron microscopy. (**a**) Gel-purified CODIs were analyzed by electron microscopy with negative staining. Typical class-averages are shown. Scale bar 20 nm. Arrows indicate the smaller particle in the pairs. (**b**) The distances between the particles within different class averages were measured using ImageJ program (version 1.54a) and plotted. Asterisks mark internucleosomal distances of 11, 15 and 18 nm formed after stepwise uncoiling of the DNA from one of the nucleosomes. (**c**) Models of dinucleosomes. The orientation of the complete (147-bp) nucleosome on the left is fixed. Major classes of dinucleosomes observed in electron microscopy likely represent structures with different linker DNA segment lengths. The distances between nucleosome cores (NCs) of 11, 15 and 18 nm (marked by 1, 2 or 3 asterisks, respectively) correspond to DNA linker lengths of −7, 4 and 15 bp, respectively, according to our model (note that visual appearance of the structures depends on the relative orientation of the two nucleosomes). The variable DNA linker length is likely observed due to different extent of DNA uncoiling from the histone octamer (Figure 2a).

**Figure 5 ijms-24-12127-f005:**
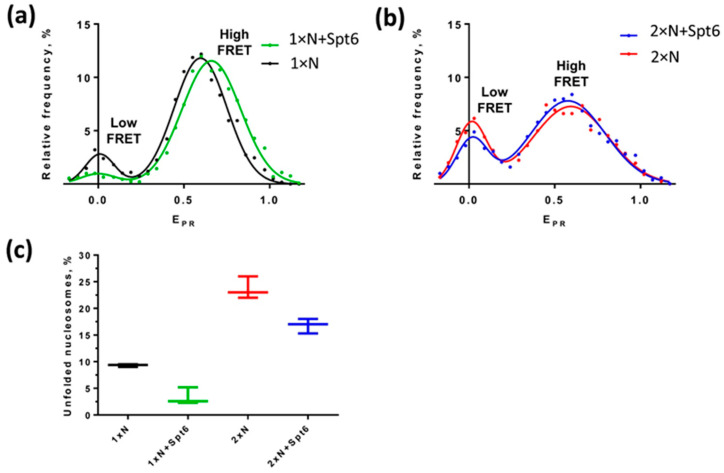
Nucleosome-binding protein Spt6 stabilizes mononucleosomes, but not dinucleosomes. Mono- (**a**) and dinucleosomes (**b**) containing the labels at the positions 13/91 bp were analyzed in the presence or in the absence of histone chaperone Spt6 using spFRET in solution. (**c**) The fraction of particles containing uncoiled DNA was calculated as the ratio (%) of an area under the low-FRET peak to the area under the whole E_PR_-profile and averaged over 3 independent experiments (mean ± SEM). Typical representative E_PR_-profiles are shown in (**a**,**b**).

## Data Availability

The data presented in this study are available on request from the corresponding author.

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
