# Peer review of "Structure and Dynamics of Compact Dinucleosomes: Analysis by Electron Microscopy and spFRET"

_ijms, 2023, doi:10.3390/ijms241512127_

Round 1

Reviewer 1 Report

The authors of the manuscript entitled " Structure and Dynamics of compact dinucleosomes: Analysis by electron microscopy and spFRET" have done a commendable job in studying the structural dynamics of compact or overlapping dinucleosomes (CODIs). The authors have presented their analysis of the structure and dynamics of CODIs containing a positioned nucleosome and an adjacent nucleosome formed on natural DNA sequence. The authors have used spFRET microscopy in solution and in gel coupled with electron microscopic studies of the purified fraction of CODIs. The authors also highlight the uncoiling of nucleosomal DNA from the surface of histone octamer facing an adjacent nucleosome with a more open structure compared to a compact mononucleosome structure. In addition the histone chaperone spt6 does not affect this relaxed nucleosome structure.

Some of my minor concerns regarding this manuscript would be as follows.

a) The authors I believe should highlight about the remodeled nucleosome (remosomes) and the remodeled dinucleosomes, and the different fractions with their 603 nucleosome positioning sequence and the natural DNA sequence template (slid nucleosomes CODIs and so on) along with naked DNA. While the authors mention about a 10% naked DNA it would be beneficial to know the percentage of the other fractions as well. In addition with respect to the addition of the fluorescent labels at positions 13/91 and 57/135bp  and 35/112bp do the authors observe species of remosomes wherein some nucleosomes have DNA pumped in or pumped out in partially remodeled samples. It would be interesting if the authors could comment on this phenomenon. The authors could additionally have a schematic representation of the same with ball pins directing the position of labels and nucleosome dyads.

2. Previous reports show the histone chaperones such as FACT boosts chromatin remodeling. The histone chaperone in itself wasn't able to remodel the nucleosomes. Could the authors also comment on their observations in this regard with regards to chromatin remodeling and histone chaperones together. Probably the  structural organization of the CODIs  and its possible role in transcription as mentioned .

Overall, the manuscript is well written and importantly suggests that spacing between nucleosome is very important both for chromatin stability and vital biological processes such as transcription and DNA repair.

Author Response

Please see the accompanying cover letter

Reviewer 2 Report

In this manuscript, the authors investigated the dynamics of compact dinucleosomes (CODI) using spFRET and electron microscopy techniques. To promote the formation of compact dinucleosomes, they employed a sequence comprising 147 bp of high-affinity 603 NPS (Nucleosome Positioning Sequence) and a natural 122 bp DNA. Mono and di-nucleosomes were purified through gel electrophoresis and utilized for spFRET and electron microscopy analyses.

The results obtained led the authors to conclude that dinucleosomes exhibit a partially flexible and uncoiled DNA configuration when compared to mono-nucleosomes. Additionally, the data from electron microscopy revealed structural heterogeneity in CODIs, possibly attributed to variations in the length of linker DNA segments.

While the experimental setup and overall study design are intriguing, it is important to note that the authors may have drawn somewhat overinterpreted conclusions based on the available data.

I have the following comments for the author’s consideration-

1.     Figures 1 and 3: The authors claim that the single bands in the gels correspond to either mono or dinucleosomes. However, it is evident from the raw data in Figure 1 that multiple bands are possible, including potential forms of both mono- and dinucleosomes. Have the authors considered this possibility? If so, the interpretation of Figures 2, 3, and 5 may need to be modified accordingly.

2.     Figure S5: It is interesting to note that the negatively stained dinucleosome sample in Figure S5 also shows the presence of mononucleosomes in the field, supporting the previous comment. The authors should discuss the possibility of mononucleosome contamination and its potential impact on the interpretation of Figure 4.

3.     Considering that cells contain a chain of nucleosomes in chromatin, it would be valuable for the authors to discuss how their dynamic model of dinucleosomes fits within the context of the cell's chromatin structure.

4.     Some remodelers, such as Isw1 and Chd1, play a crucial role in resolving dinucleosomes in chromatin. This contradicts the proposed hypothesis. The authors should address this discrepancy and provide a discussion around it.

Minor points:

5.     It would be beneficial for the authors to include the data for mononucleosomes from electron microscopy, similar to what is shown in Figure 4 and Figure S5.

6.     Please define the term "spFRET" the first time it is used in the article.

7.     In line 110, where it says "...Cy5 (red circle)...," it should be corrected to "...Cy5 (green circle)..." to accurately represent the color of the circle.

Author Response

(The authors gave the same response as above.)

Round 2

Reviewer 2 Report

The authors have addressed all my comments. 

The article is well-written and will be interesting for the readers from chromatin field.